# Hydrogeochemical Study of Hot Springs along the Tingri—Nyima Rift: Relationship between Fluids and Earthquakes

Deyang Zhao [1], Xiaocheng Zhou [2,3,*], Yongxian Zhang [2,*], Miao He [2], Jiao Tian [2], Junfeng Shen [3], Ying Li [2,3], Guilan Qiu [1], Fang Du [1], Xiaoming Zhang [1], Yao Yang [1], Jun Zeng [1], Xuelian Rui [1], Feng Liao [1] and Zhijun Guan [1]

1   Sichuan Earthquake Administration, Chengdu 610041, China
2   United Laboratory of High-Pressure Physics and Earthquake Science, Institute of Earthquake Forecasting, China Earthquake Administration, Beijing 100036, China
3   School of Earth Sciences and Resources, China University of Geosciences, Beijing 100083, China
*   Correspondence: zhouxiaocheng188@163.com (X.Z.); yxzhseis@sina.com (Y.Z.)

**Abstract:** Studying the hydrogeochemical characteristics of hot springs provides essential geochemical information for monitoring earthquake precursors and understanding the relationship between fluids, fractures, and earthquakes. This paper investigates the hydrogeochemical characteristics of hot springs along the Tingri–Nyima Rift (TNR) in southern Tibet, a seismically active zone at the collision front of the Indian and Asian-European plates. The major elements, hydrogen, and oxygen isotopes of seven thermal springs were analyzed from July 2019 to September 2021. The findings indicate that Mount Everest's meteoric water, which has a recharge elevation of roughly 7.5–8.4 km, is the main source of recharge for the hot springs. The water samples have two main hydrochemical types: $HCO_3$-Na and Cl-Na. The temperature of the geothermal reservoir is between 46.5 and 225.4 °C, while the circulation depth is between 1.2 and 5.0 km based on silica-enthalpy mixing models and traditional geothermometers. Furthermore, continuous measurements of major anions and cations at the Yundong Spring (T06) near Mount Everest reveal short-term (8 days) seismic precursor anomalies of hydrochemical compositions before an $M_L$ 4.7 earthquake 64.36 km away from T06. Our study suggests that seismicity in the northern section of the TNR is controlled by both hydrothermal activity and tectonic activity, while seismicity in the southern section is mainly influenced by tectonic activity. In addition to magnitude and distance from the epicenter, geological forces from deep, large fissures also affect how hot springs react to seismic occurrences. A fluid circulation model is established in order to explain the process of groundwater circulation migration. The continuous hydrochemical monitoring of hot springs near Everest is critical for studying the coupling between hot springs, fractures, and earthquakes, as well as monitoring information on earthquake precursory anomalies near Everest.

**Keywords:** hot spring; isotopes; hydrogeochemistry; seismic precursor; Tingri–Nyima Rift; Mount Everest

## 1. Introduction

Fluids in the crust play a crucial role in material transport and energy exchange along seismically active fault zones. The interaction between fluids and rocks can also affect the activity and properties of the fault zone [1,2]. Therefore, studying the hydrogeological characteristics of hot springs along active faults can provide valuable information on the source and movement of geothermal fluids [3–5].

The Qinghai–Tibet Plateau is part of the Mediterranean-Himalayan geothermal belt that formed due to the closure of the Tethys Ocean and the collision of the Indo-Asian plate. It has strong hydrothermal activity [6,7]. Previous research on hot springs in southern Tibet mainly focused on geothermal resource evaluation [8,9] and deep crustal and mantle

structures [10–12]. However, most studies were focused on three rift regions: Cuona–Voka Rift, Yadong–Gulu Rift, and Dingjie–Shenza Rift (Figure 1b). The hydrogeochemical characteristics of hot springs along the Tingri–Nyima Rift (TNR) have been poorly reported. Moreover, after the Nepal *M*8.0 earthquake in 2015, seismic risk increased significantly in this region [13–16], but seismic monitoring stations are lacking and the coupling relationship between hot springs, faults, and earthquakes is poorly reported. In this paper, we aim to fill these gaps by investigating the hydrochemical properties and origin of seven hot springs along the TNR. We analyzed major elements, trace elements, and δD and δ¹⁸O values to characterize their hydrochemical types and sources. We also calculated their reservoir temperatures and circulation depths using common geothermometers and silica-enthalpy mixing models to reveal the possible hydrochemical evolution. Furthermore, we monitored the T06 hot spring near Mount Everest continuously, starting in September 2021, to detect any seismic precursor anomalies before nearby earthquakes. We aimed to investigate the coupling relationship between faults, fluids, and earthquakes with a theoretical model of fluid circulation and study the changes in the ion concentrations of hot springs for seismic precursor anomalies.

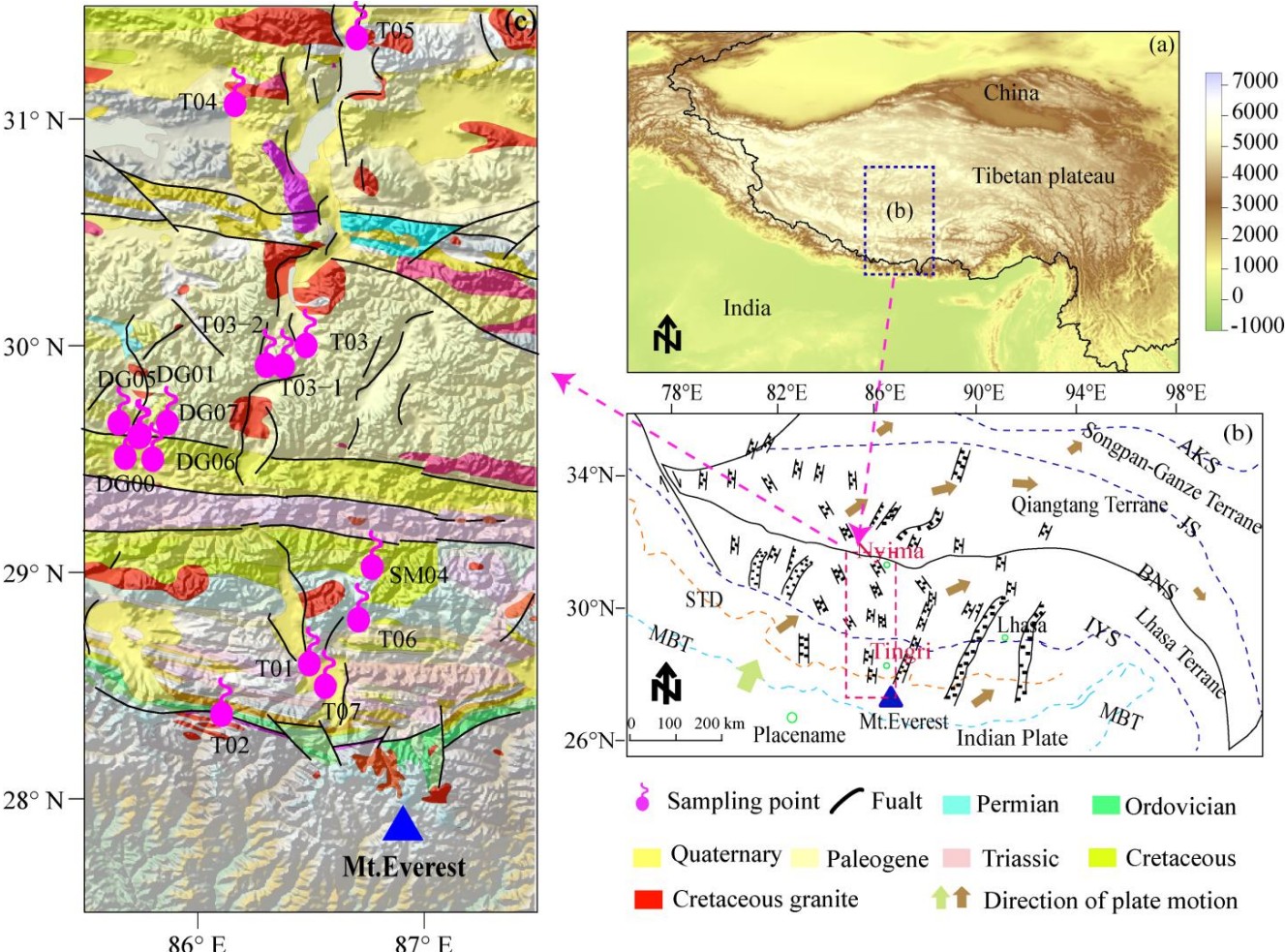

**Figure 1.** Distribution plot of sample sites: (**a**) Geographic location of the study area; (**b**) simplified map showing the tectonic framework of Tibet with N−S striking rifts adapted from Wang et al. [9]; Jinshajiang Suture Zone (JS), South Tibet Detachment (STD), Main Border Thrust Fault (MBT); (**c**) geological map and topographic map in the TNR.

## 2. Geological Setting

The South Tibetan Rift System (STRS) is a prominent active tectonic structure within the Tibetan Plateau that formed due to the collision of the Indian and Eurasian plates. It consists of several north–south rift valleys distributed between the Karakorum–Jiali rift system in the north and the Himalayan orogenic belt in the south (82°–92°E), with each rift valley spaced about 150–200 km apart. The STRS reflects the present-day near east–west extension of the Tibetan Plateau [17,18]. The study area, the Tingri–Nyima Rift (TNR), is situated in the center of STRS (Figure 1b). TNR is divided into two parts, north, and south, by the Indus-Yarlung Suture Zone (IYS), which extends about 380 km north–south and 25 km east–west. South of IYS, the Kung Co rift in the Himalaya block is a long normal fault that dips steeply to the west (50°–60°), with an estimated rift initiation of ~12–13 Ma and normal fault slip rate acceleration of ~10 Ma [19]. The Tangra Yumco rift in the Lhasa block is a dextral strike-slip fault from IYS to near the Bangong suture (BNS). Normal faulting of the Tangra Yumco segment began at about $14.5 \pm 1.8$ Ma [20]. Due to highly intense active tectonics, the TNR zone has become one of the most seismically active areas. According to data released by The China Earthquake Network Center, since 1833 there have been nine earthquakes with $M \geq 6$, two earthquakes with $M \geq 7$, and one earthquake with $M \geq 8$.

The regional stratigraphy mainly includes Triassic–Cretaceous volcanic sedimentary rocks, Paleogene and Cretaceous igneous rocks (especially granite), and a small amount of Ordovician, Silurian, Devonian, Triassic limestone, and carboniferous Permian metasedimentary rocks. In the process of the Indian plate moving northward and colliding with the Eurasian plate, the active block south of the BNS zone exhibits a high heat flow [21], ranging from 60–364 $mW/m^2$, with an average value greater than 100 $mW/m^2$, which is caused by factors such as faults, collisional orogeny, and the shear heat generation of mantle upwelling. Along the TNR, a greater number of geothermal springs have been examined, the origins of which include deep magma heat, radioactive heat from granitoid, and strike-slip frictional heat.

## 3. Sampling and Methods

In this investigation, groundwater samples were taken from 7 hot spring locations along the TNR (T01–T07) in different periods (Figure 1b and Table 1). Water samples were gathered in five colorless polyethylene terephthalate bottles (50 mL) and filtered through a 0.45 μm membrane before being analyzed for the content of major and trace elements, hydrogen and oxygen isotopes, and silicon dioxide. On-site water temperatures were measured using a digital thermometer with 0.1 °C precision. The pH and conductivity parameters were measured on the spot using handheld meters that had been calibrated before sampling. By using Dionex ICS-900 ion chromatograph and AS40 automated sampler (Thermo Fisher Scientific, Waltham, MA, USA) with a 5% or less repeatability and a 0.01 mg/L detection limit in the Earthquake Forecasting Key Lab of China Earthquake Administration [22], the principal elements were determined. A ZDJ-100 potentiometer titrator (INASE Scientific Instrument CO., LTD., Shanghai, China) technique was used to determine the amounts of $HCO_3^{-}$ and $CO_3^{2-}$. The procedure had a repeatability of 2% and used 0.05 mol/L HCl with 0.1% methyl orange and 1% phenolphthalein. When expressed as (cations anions)/(cations + anions), the ionic charge equilibrium specified in meq/L was within 10% for samples. The hydrogen and oxygen isotopes were evaluated by the TC/EA technique on a Finnigan MAT253 mass spectrometer (Thermo Fisher Scientific, MA, USA). The isotope accuracy values for the V-SMOW and the studied water samples were 0.2% and 1%, respectively [23]. The $SiO_2$ content of the samples was assessed using the Optima-5300 DV(PerkinElmer, MA, USA) [24].

**Table 1.** Location of the surveyed area of hot springs in the TNR.

| No. | Date (dd/mm/yyyy) | Site | Longitude (°) | Latitude (°) | Altitude (m) | Style | References |
|---|---|---|---|---|---|---|---|
| T01 | 31 July 2019 | Canmuda | 86.49 | 28.6 | 4365 | | |
| T02 | 1 August 2019 | Yadang | 86.1 | 28.38 | 4331 | | |
| T03 | 3 October 2020 | Chazi | 86.48 | 30 | 4841 | | |
| T04 | 4 October 2020 | Bianla | 86.16 | 31.07 | 4772 | Spring water | this study |
| T05 | 4 October 2020 | Maerzuo | 86.7 | 31.47 | 4573 | | |
| T06 | 22 September 2021 | Yundong | 86.71 | 28.79 | 4649 | | |
| T07 | 25 September 2021 | Saba | 86.56 | 28.5 | 4518 | | |
| T03-1 | - | Chazi | 86.48 | 30 | 4841 | geothermal water | |
| T03-2 | - | Chazi | 86.48 | 30 | 4837 | | LUO et al. [25] |
| T03-3 | - | Chazi | 86.48 | 30 | 4848 | river water | |
| NW01 | 13 August 2013 | Nima | 87.23 | 31.79 | 4553 | | |
| NW02 | 15 August 2013 | Asuo Town | 86.06 | 31.88 | 4797 | | |
| NW03 | 15 August 2013 | Juncang Town | 86.01 | 31.25 | 4726 | natural water | Tian et al. [26] |
| NW04 | 22 August 2013 | Sangsang Town | 86.72 | 29.42 | 4632 | | |
| DG00 | - | Daggyai | 85.75 | 29.5 | 4974 | | |
| DG01 | - | Daggyai | 85.75 | 29.5 | 4974 | | |
| DG05 | - | Daggyai | 85.75 | 29.52 | 5012 | geothermal water | |
| DG06 | - | Daggyai | 85.75 | 29.52 | 5012 | | |
| DG07 | - | Daggyai | 85.75 | 29.52 | 4974 | | Liu et al. [27] |
| DG-R2 | - | Daggyai | 85.75 | 29.5 | 4974 | river water | |
| SM04 | - | Semi | 86.77 | 29.02 | 5051 | geothermal water | |

Note: "-" represent no data.

## 4. Results and Discussion

Table 2 displays the findings of the analysis performed on the water samples from hot springs. All samples had ion balances that were less than 10%, suggesting that the conclusions drawn from the analysis of these samples were reasonable. Water springs had temperatures between 7.5 °C and 87 °C and a pH between 6.79 and 8.56. TDS readings varied from 105 to 4865.9 mg/L, while conductivity ranged from 211 to 7090 μs/cm. $Na^+$, $K^+$, $Ca^{2+}$ and $Mg^{2+}$ were the major cations, whereas $Cl^-$, $SO_4^{2-}$, and $HCO_3^-$ were the major anions in hot springs. $Na^+$, $K^+$, $Ca^{2+}$ and $Mg^{2+}$ concentrations varied from 12.00 to 1460.79 mg/L, 1.39 to 185.58 mg/L, 9.51 to 1164.68 mg/L, and 0.00 to 277.84 mg/L, respectively; $Cl^-$, $SO_4^{2-}$, and $HCO_3^-$ concentrations varied from 1.21 to 1019.63 mg/L, 2.73 to 895.45 mg/L, and 106.60 to 2222.2 mg/L, respectively.

### 4.1. Origin of Hot Spring Water

This study analyzed the hydrological cycle of groundwater and recharge by comparing the $\delta^{18}O$ and $\delta D$ from samples with the Global Meteoric Water Line ($\delta D = 8.0 \times {}^{18}O + 10$) [28]. The oxygen and hydrogen isotopic compositions from the hot springs ranged from −169.50‰ to −134.90‰ and from −20.90‰ to −11.80‰, respectively. Figure 2 shows the $\delta D$–$\delta^{18}O$ relationship diagram for the spring water, geothermal field water, river, and snowmelt water. The local meteoric water line (LMWL) is very close to the Global Meteoric Water Line [29], and the isotope concentrations of snowmelt are based on the work of Guo et al. [30]. Moreover, the red rectangle area also demonstrates the graph of magmatic water, with $\delta D$ of −20 ± 10‰ and $\delta^{18}O$ of 10 ± 2‰ [31].

**Table 2.** Spring samples' chemical composition along the TNR.

| No. | Temperature | PH | EC | TDS | Li | Na$^+$ | K$^+$ | Mg$^{2+}$ | Ca$^{2+}$ | F$^-$ | Cl$^-$ | Br | SO$_4{}^{2-}$ | NO$_3{}^-$ | CO$_3{}^{2-}$ | HCO$_3{}^-$ | Si | Water Types | δD | δ$^{18}$O | Recharge Elevation |
|---|---|---|---|---|---|---|---|---|---|---|---|---|---|---|---|---|---|---|---|---|---|
| | °C | | µs/cm | | | | | | mg/L | | | | | | | | | | | ‰ | km |
| T01 | 44 | 6.79 | 2168 | 1313.6 | 1.73 | 325.04 | 59.86 | 25.95 | 128.14 | 4 | 18.04 | 0 | 2.73 | 5.42 | 0 | 2060 | 16.9 | HCO$_3$-Na•Ca | −156 | −19.2 | 7.6 |
| T02 | 19.2 | 6.84 | 1748 | 1000.5 | 0.1 | 17.33 | 6.31 | 277.84 | 31.51 | 1.29 | 1.21 | 0.13 | 66.3 | 5.1 | 0 | 1646 | 5.76 | HCO$_3$-Mg | −154.6 | −17.9 | 7.5 |
| T03 | 73.5 | 8.16 | 2121 | 1309.2 | 2.26 | 458.8 | 42.96 | 1.34 | 17.71 | 19.43 | 179.33 | 25.92 | 126.66 | 0 | 59 | 1135 | 53.2 | HCO$_3$-Na | −169.5 | −20.3 | 8.4 |
| T04 | 57 | 7.4 | 3610 | 2437.3 | 1.08 | 797.76 | 47.87 | 7.88 | 55.2 | 8.35 | 143.51 | 2.51 | 577.93 | 0 | 0 | 2180 | 28.5 | HCO$_3$•SO$_4$-Na | −166.6 | −20.9 | 8.2 |
| T05 | 20 | 7.11 | 7090 | 4865.9 | 8.59 | 1460.79 | 185.58 | 16.49 | 152.92 | 5.71 | 1019.63 | 6.6 | 895.45 | 3.03 | 0 | 3046 | 29.7 | HCO$_3$•Cl-Na | −148.2 | −19.2 | 7.6 |
| T06 | 63.8 | 7.26 | 1836 | 1135.12 | 2.88 | 360.59 | 30.78 | 6.88 | 65.02 | 8.3 | 71.48 | 0.19 | 0 | 15.11 | 0 | 1147.78 | 20.7 | HCO$_3$-Na | −162.7 | −20.8 | 8.0 |
| T07 | 28.4 | 7.07 | 3268 | 2104.87 | 4.37 | 627.14 | 97.18 | 12.08 | 164.68 | 6 | 80.72 | 0.26 | 3.26 | 0.57 | 0 | 2217.23 | 14.3 | HCO$_3$-Na | −164.8 | −20.9 | 7.9 |
| T03-1 | 87 | 7.83 | - | 1806.97 | - | 481.9 | 39.12 | <0.10 | 12.77 | - | 165.05 | - | 121.94 | - | - | 654.94 | 64 | HCO$_3$•Cl-Na | −163 | −20.5 | 8.2 |
| T03-2 | 74 | 7.93 | - | 1844.14 | - | 476.91 | 41.7 | <0.10 | 9.51 | - | 167.77 | - | 117.52 | - | - | 707.28 | 71.05 | HCO$_3$•Cl-Na | −163 | −20.6 | 8.2 |
| T03-3 | 7.5 | 7.48 | - | 278.98 | - | 13.05 | 1.39 | 6.6 | 40.88 | - | 1.41 | - | 47.26 | - | - | 115.96 | | HCO$_3$•SO4-Ca | −163 | −17.6 | |
| NW01 | - | 7.89 | 669 | 334 | - | 98.1 | 5.6 | 25 | 26.1 | - | 50.7 | - | 64.6 | - | - | 195.3 | 6.39 | HCO$_3$-Na•Mg | - | - | - |
| NW02 | - | 7.84 | 540 | 270 | - | 45.2 | 6.5 | 19.8 | 56.8 | - | 21.6 | - | 49.5 | - | - | 286.8 | 5.41 | HCO$_3$-Ca•Na•Mg | - | - | - |
| NW03 | - | 7.89 | 340 | 170 | - | 16.4 | 1.8 | 5.3 | 59.5 | - | 39.9 | - | 9.5 | - | - | 146.4 | 7.09 | HCO$_3$•Cl-Ca | - | - | - |
| NW04 | - | 7.55 | 457 | 229 | - | 12 | 1.2 | 14.1 | 69.8 | - | 61.9 | - | 36.2 | - | - | 112.9 | 7.28 | HCO$_3$•Cl-Ca | - | - | - |
| DG00 | 79.3 | 8.47 | 1873 | 918 | - | 453.9 | 52.9 | 1.3 | 43.1 | 24.5 | 155.5 | - | 85.5 | - | - | 393.5 | 137.9 | HCO$_3$•Cl-Na | −153.3 | −20.4 | 8.1 |
| DG01 | 79.5 | 8.16 | 1910 | 935 | - | 440.1 | 52.1 | 1.3 | 39.4 | 24.1 | 153.5 | - | 83.4 | - | - | 506.2 | 136.8 | HCO$_3$•Cl-Na | −153.2 | −20.1 | 8.1 |
| DG05 | 78.9 | 8.44 | 2004 | 982 | - | 482.4 | 56.9 | 0.7 | 33.9 | 26.3 | 162.3 | - | 90.1 | - | - | 536 | 167.7 | HCO$_3$•Cl-Na | −147 | −18.7 | 8.0 |
| DG06 | 74.1 | 8.56 | 1916 | 939 | - | 432.3 | 53.7 | 0.9 | 26 | 24.7 | 152 | - | 87 | - | - | 670.6 | 149.8 | HCO$_3$•Cl-Na | −147.6 | −20.8 | 8.0 |
| DG07 | 82.1 | 8.28 | 1872 | 909 | - | 451.2 | 53 | 0.8 | 23.5 | 24.1 | 148.7 | - | 88 | - | - | 697.5 | 151.6 | HCO$_3$-Na | −147.9 | −19.6 | 8.0 |
| DG-R2 | 12.2 | 7.62 | 211 | 105 | - | 62.3 | 10.1 | 2.9 | 52.9 | 1.8 | 10.2 | - | 19.5 | - | - | 106.6 | 14.9 | HCO$_3$-Na•Mg | −134.9 | −18.1 | 7.7 |
| SM04 | 84.9 | 8.46 | 4184 | 2050 | - | 945.5 | 161.8 | 1.4 | 15.8 | 19 | 751.8 | - | 29.2 | - | - | 188.8 | 232.5 | Cl-Na | −143.2 | −11.8 | 8.0 |

Note: "-"represent no data.

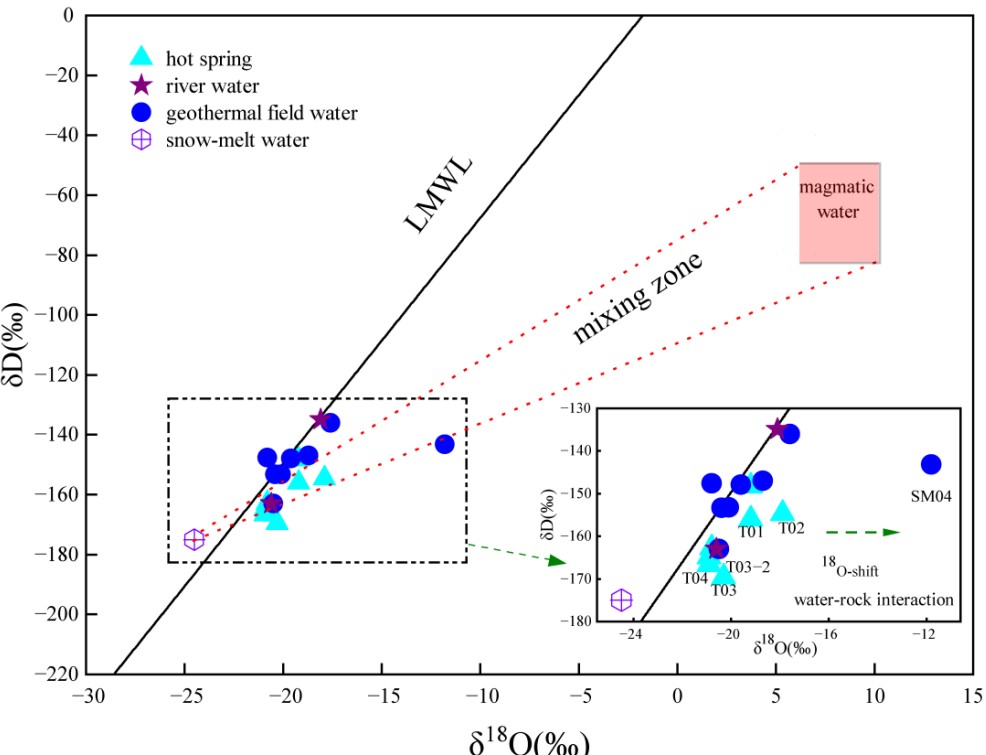

**Figure 2.** $\delta D$–$\delta^{18}O$ plot from the hot springs, geothermal field waters, river waters, and snow-melting water in South Tibet. The LMWL ($\delta D = 8.0 \times \delta^{18}O + 14.4$) and magmatic fluid distribution ($\delta^{18}O = 7 \pm 2‰$, $\delta D = -25 \pm 10‰$) are from Tan et al. [27].

The majority of the samples in Figure 2 fall around the atmospheric precipitation line, showing that they are mostly the result of atmospheric precipitation infiltration. The hot spring water samples fall within the mixing area and exhibit strong $\delta^{18}O$ enrichment, which may be related to the evaporation effect. Some samples showed different degrees of oxygen shift, and they also fell within the line between snowmelt water and magma water (mixing zone), especially the SM04 sample. We can speculate that magmatic fluids might also participate in the formation process of geothermal fluids based on the water-bearing granitic magma melt at depth this region [29,32]. Additionally, the geothermal fluid reacts with the surrounding rock during the rising process, and more oxygen isotopes are obtained.

The phenomenon in which the contents of $\delta D$ and $\delta^{18}O$ decrease with the height increase in atmospheric precipitation is called the isotope elevation effect. This effect can be applied to estimate the recharge elevation of water samples [33]. The recharge elevation formula of the $\delta D$ value in precipitation and local altitude is H = h+(R-R*)/K, where h is sampling point elevation; R is the $\delta D$ value from the sampling water; K is height gradient of the $\delta D$ value, which is about $-4.2‰$ km$^{-1}$ [34]; and R* is the $\delta D$ value from atmospheric precipitation in the study area, which is $-20.85‰$. Thus, the recharging elevation was estimated to be approximately 7.5–8.4 km, suggesting that the hot spring water may be recharged by snowmelt from Mount Everest.

### 4.2. Origin of Water-Soluble Ions
#### 4.2.1. Origin of Major Elements

Figure 3 displays the results of Piper triad mapping of the chemical composition of the study area using Origin software. Based on the Shukalev classification, we can categorize most of the groundwater samples into nine chemical types (Table 2): $HCO_3 \bullet SO_4$-Ca, $HCO_3$-Ca$\bullet$Na$\bullet$Mg, $HCO_3 \bullet Cl$-Ca, $HCO_3$-Na$\bullet$Mg, $HCO_3$-Mg, $HCO_3$-Na$\bullet$Ca, $HCO_3$-Na, $HCO_3 \bullet Cl$-Na, Cl-Na. Our analysis reveals that Ca$^+$ was the main cation in river water and

cold spring samples, and $Na^+$ was the main cation in hot spring and geothermal water samples. $HCO^{3-}$ was the main anion in all samples (with $Cl^-$ being the main anion in SM04 samples). Interestingly, Figure 2 shows a gradual reduction in $Na^+$ content proportion in groundwater samples from SM04 to other hot spring samples, geothermal water samples, and river water samples. This finding suggests that hydrochemical types of water samples can help to reveal the origin of major ions [35]. The presence of high concentrations of Na-HCO₃ in groundwater typically signifies $CO_2$-involved water–rock interaction at great depth. This suggests that the origin of the Na-HCO₃-type waters in the geothermal field is likely linked to the dissolution of Na/K-silicates such as albite and K-feldspar in the reservoir rocks [36]. We can represent this process using Equations (1) and (2). As for neutral Cl-Na water, it is commonly believed to be directly formed by deep geothermal fluid through both "chemical dissolution" and "hydrothermal alteration mineral formation" processes that are often related to deep magmatic activity. This type of water is commonly found in hot springs, boiling springs, and hot water with high temperatures [37,38].

$$2NaAlSi_3O_8 + 9H_2O + 2H_2CO_3 \rightarrow Al_2Si_2O_5(OH)_4 + 4H_4SiO_4 + 2Na^+ + 2HCO_3^- \quad (1)$$

$$2KAlSi_3O_8 + 3H_2O + 2CO_2 \rightarrow Al_2Si_2O_5(OH)_4 + 4SiO_2 + 2K^+ + 2HCO_3^- \quad (2)$$

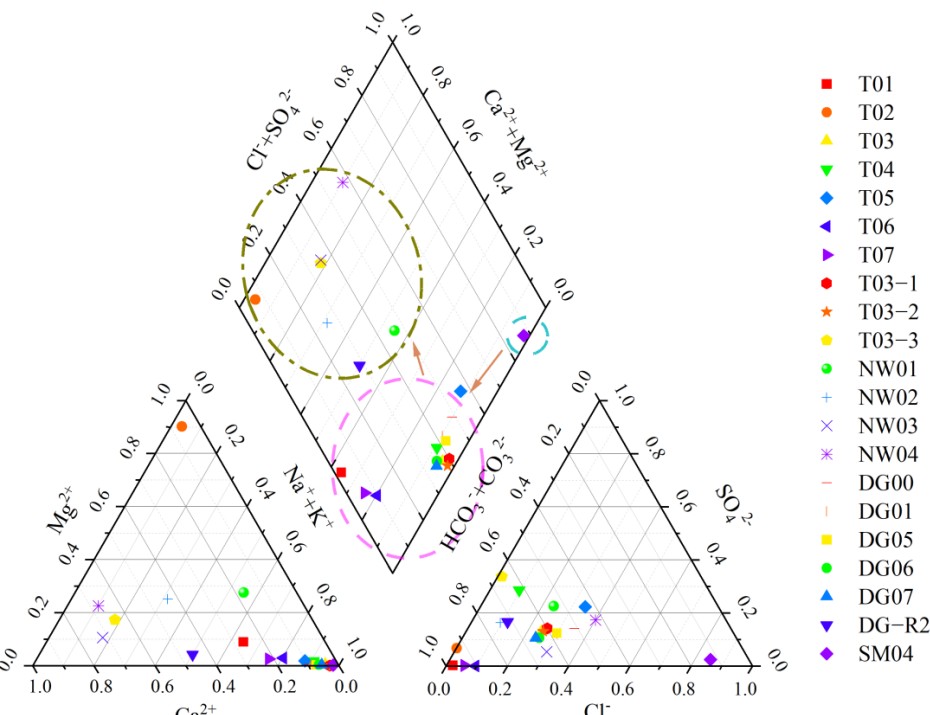

**Figure 3.** Piper diagram of water samples along the TNR.

4.2.2. Origin of Trace Elements

The trace element contents in hot springs are influenced by regional geochemical background values, element properties, water–rock reaction degree, and groundwater mixing from different sources [39,40]. Typically, the Enrichment Factor (EF) method is employed to determine the enrichment levels of elements [41]. The calculation method for EF is expressed by Equation (3).

$$EF_i = \left(\frac{C_i}{C_R}\right)_W / \left(\frac{C_i}{C_R}\right)_r \quad (3)$$

where $C_i$ is the content of the selected reference element; $C_R$ is the element content in the sample; w is the concentration of elements in the water sample; r is the concentration of

elements in the rock. The upper crust rocks were selected as the reference background value in this paper [42], and Al in the crustal elements was selected as the reference element [43]. The 21 trace elements measured in the 7 hot springs were normalized, and the normalized $EF_i$ is shown in Figure 4. As can be seen from the data in Figure 4, Li, B, and Sr have relatively high concentrations. Metal from the alkaline earth, Sr is a dispersive element that is quite abundant in the crust and mantle. It was easier to enrich in fluids that were only slightly alkaline, and its migration is typically linked to calcium [44,45]. In this study area, potassium feldspar and hornblende-rich volcanic and granitic rocks have formed. The spring water samples in the TNR had a mean pH of 7.76 and a pH range of 6.79 to 8.56, which was conducive to Sr enrichment. Li has active chemical properties and a strong mobility and is primarily enriched in acidic rocks with a high migration capacity. In hot springs in the Tibetan region, the high concentration of Li is largely derived from lithium silicate minerals present in granite and granodiorite, which may be due to the upwelling of deeper fluids [46,47]. In addition, previous studies have shown that the solubility of B in groundwater increases with depth and pressure. Thus, the deeper the groundwater circulation, the higher the B content [27,44]. The relative enrichment of Li, B, and Sr elements in the area indicates deep circulation.

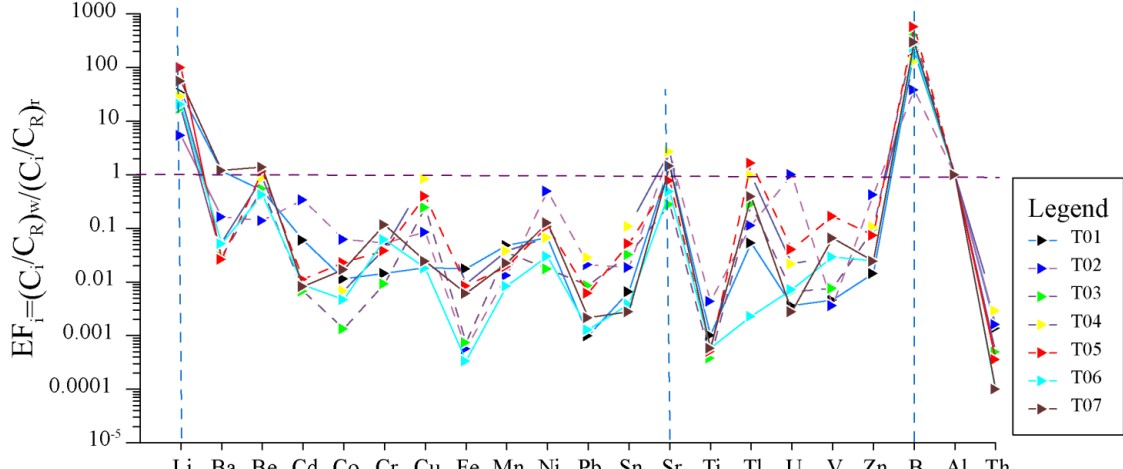

**Figure 4.** Enrichment factors of trace elements in hot springs along the TNR.

*4.3. Water–Rock Interaction of Hot Springs*

4.3.1. The Water–Rock Reaction Equilibrium

The Na-K-Mg ternary diagram is a popular tool used to evaluate the water–rock equilibrium state of hot water, classify different types of water samples, and estimate the equilibrium temperature [48]. The Na-K-Mg triangle diagram (Figure 5) shows that each sample is not fully equilibrated. River water samples lie close to the Mg end-member, whereas hot spring waters are in the immature water zone, and several geothermal waters are in the partially equilibrated zone. This result suggests that geothermal water mixes with shallow cold water as it ascends towards the surface [49]. From the data presented in Figure 5, it's apparent that T03-1 and T03-2 are positioned closer to the equilibrium line, with reservoir temperatures of about 200–220 °C, which are similar to the results obtained from the Na-K geothermometer (Table 3).

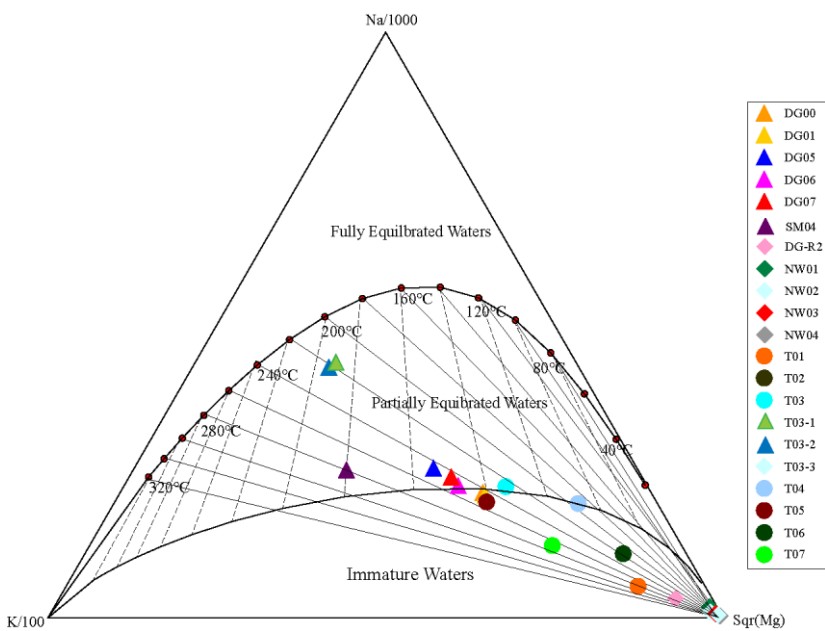

**Figure 5.** Na-K-Mg ternary diagram of samples along the TNR.

### 4.3.2. Reservoir Temperature and Circulation Depth

To estimate reservoir temperatures, a range of chemical geothermometers, such as cationic, silica, gas, and isotope geothermometers, are typically utilized [48,50–53]. Yet, due to the intricate geologic context, various geothermometers frequently produce extremely varied reservoir temperature estimations. Therefore, this study employs multiple geothermometers, as shown in Table 4, to optimize the results and find the most suitable geothermometer. The results indicate that the Na-K geothermometer yields the highest reservoir temperature estimate (173.21–360.19 °C) compared to other geothermometers. However, the results from other geothermometers vary greatly, and further confirmation using other methods is required.

This work used the silica-enthalpy mixing models proposed by Fournier to analyze the mixture of deeper hot water and shallower cold water, and compute the cold water mixing ratio and deep thermal storage temperature. [51]. The extrapolations of the lines from the cold spring water sample (T02) to the data points of the Daggyai geothermal water samples and Semi geothermal water samples do not cross the quartz solubility curve, showing that these water samples mixed minimally or not at all with shallower cold water while ascending to the surface in the area [9]. The intersections (A and B) of the line between the samples and the steam point and the quartz solubility curve determine the enthalpy and temperature of deep geothermal water. The reservoir temperatures at points A and B are 226–238 °C and 175–200 °C, respectively, which are close to the results calculated by the quartz geothermometer with the maximum steam loss. For other water sample points, the extrapolations of the lines from the cold spring water sample (T02) to hot spring water samples also show that points A and B intersect with the quartz solubility curve [54]. Meanwhile, the temperature of the reservoir corresponding to the intersection of C and D is approximately 134 °C and 123 °C, respectively, when the hot water steam loss is at its greatest before mixing. From Figure 6, it can be calculated that the mixing ratio of cold water is about 65–90%.

**Table 3.** The concentrations of trace elements in the spring waters along the TNR (µg/L).

| No. | Ag | Al | Ba | Be | Cd | Co | Cr | Cu | Fe | Mn | Mo | Ni | Pb | Sb | Sn | Sr | Th | Ti | Tl | U | V | Zn | B |
|-----|------|------|------|-------|--------|-------|------|------|------|------|-------|------|-------|-------|-------|------|-------|------|-------|-------|-------|------|-------|
| T01 | 0.004 | 19 | 1526 | 3.61 | 0.014 | 0.271 | 1.2 | 1.09 | 1463 | 68.6 | 0.096 | 3.16 | 0.047 | 0.22 | 0.086 | 1301 | 0.035 | 7.22 | 0.095 | 0.024 | 0.655 | 2.43 | 7099 |
| T02 | 0.011 | 6.49 | 73 | 0.339 | 0.027 | 0.505 | 1.51 | 1.72 | 16.1 | 6.45 | 0.152 | 8.02 | 0.341 | 0.455 | 0.083 | 742 | 0.014 | 10.6 | 0.069 | 2.28 | 0.176 | 24.4 | 466 |
| T03 | 0.007 | 48.5 | 99 | 9.99 | 0.004 | 0.081 | 1.96 | 37.1 | 157 | 129 | 0.248 | 2.13 | 1.04 | 17.3 | 1.08 | 591 | 0.032 | 6.88 | 1.28 | 0.111 | 2.76 | 41.2 | 36841 |
| T04 | 0.004 | 13.6 | 50.9 | 4.46 | <0.002 | 0.116 | 2.47 | 35.6 | 545 | 38.7 | 83.9 | 2.28 | 0.973 | 1742 | 1.02 | 1599 | 0.053 | 2.73 | 1.24 | 0.103 | 3.08 | 12.9 | 3292 |
| T05 | 0.011 | 29.2 | 53.1 | 12.7 | 0.004 | 0.837 | 4.89 | 36.3 | 1078 | 42.3 | 0.758 | 7.89 | 0.451 | 22.9 | 1.04 | 991 | 0.014 | 5.51 | 4.53 | 0.413 | 36.8 | 19.2 | 31593 |
| T06 | 0.026 | 56 | 199 | 8.89 | 0.006 | 0.327 | 15.1 | 3.16 | 82.2 | 35.1 | 0.041 | 4.2 | 0.18 | 0.089 | 0.148 | 1197 | 0.007 | 12.2 | 0.012 | 0.141 | 12.3 | 12.2 | 24929 |
| T07 | 0.011 | 29.8 | 2466 | 15.5 | 0.003 | 0.634 | 15.2 | 2.27 | 794 | 50.1 | 0.105 | 9.45 | 0.16 | 0.022 | 0.057 | 1899 | 0.004 | 6.48 | 1.1 | 0.029 | 14.7 | 6.44 | 16616 |

**Table 4.** Reservoir temperature and circulation depth of the samples along the TNR.

| No. | Na-K (°C) | K-Mg (°C) | Na-K-Ca (°C) | Cdy (°C) | Cdy,msl (°C) | Qz (°C) | Qz,msl (°C) | SEM (°C) | SEM, msl (°C) | CD (km) |
|---|---|---|---|---|---|---|---|---|---|---|
| T01 | 275.59 | 100.27 | 137.85 | 56.44 | 63.82 | 87.34 | 90.00 | 162.00–167.00 | 113.00 | 2.01 |
| T02 | 360.19 | 24.96 | 56.10 | 13.63 | 26.48 | 46.23 | 53.56 | - | - | 1.20 |
| T03 | 211.41 | 134.73 | 193.62 | 118.78 | 115.40 | 144.65 | 138.97 | 240.00–243.00 | 145.00 | 3.10 |
| T04 | 176.78 | 110.78 | 168.33 | 82.20 | 85.52 | 111.38 | 110.79 | 175.00–200.00 | 123.00 | 2.47 |
| T05 | 238.40 | 141.71 | 218.55 | 84.41 | 87.35 | 113.41 | 112.53 | - | - | 2.51 |
| T06 | 203.76 | 100.22 | 206.38 | 65.98 | 71.92 | 96.31 | 97.79 | 135.00 | | 2.18 |
| T07 | 257.65 | 125.69 | 272.57 | 48.98 | 57.42 | 80.28 | 83.82 | 240.00–243.00 | 145.00 | 1.87 |
| T03-1 | 199.74 | 178.45 | 201.84 | 131.18 | 125.29 | 155.70 | 148.18 | 240.00–243.00 | 145.00 | 3.30 |
| T03-2 | 205.74 | 181.03 | 217.64 | 138.43 | 131.02 | 162.11 | 153.49 | 240.00–243.00 | 145.00 | 3.42 |
| DG00 | 230.47 | 142.22 | 172.32 | 191.83 | 172.01 | 208.20 | 190.95 | 175.00–200.00 | - | 4.25 |
| DG01 | 231.89 | 141.70 | 174.12 | 191.10 | 171.46 | 207.58 | 190.46 | 226.00–238.00 | - | 4.24 |
| DG05 | 231.56 | 155.64 | 185.98 | 210.34 | 185.74 | 223.72 | 203.30 | 226.00–238.00 | - | 4.53 |
| DG06 | 236.31 | 149.09 | 190.61 | 199.49 | 177.72 | 214.65 | 196.10 | 226.00–238.00 | - | 4.37 |
| DG07 | 231.18 | 150.70 | 194.53 | 200.61 | 178.55 | 215.59 | 196.85 | 226.00–238.00 | - | 4.38 |
| SM04 | 267.82 | 182.51 | 311.95 | 244.77 | 210.66 | 252.01 | 225.44 | 226.00–238.00 | - | 5.02 |

Notes: Qz = quartz geothermometer with no loss; Qz,msl = quartz geothermometer with maximum steam loss; Cdy = chalcedony geothermometer with no loss; Cdy,msl = chalcedony geothermometer with maximum steam loss; Na-K = Na-K geothermometer; K-Mg = K-Mg geothermometer; Na-K-Ca = Na-K-Ca geothermometer; SEM = silica-enthalpy mixing models; SEM,msl = maximum-steam-loss SEM; CD = circulation depth; "-" represents no data.

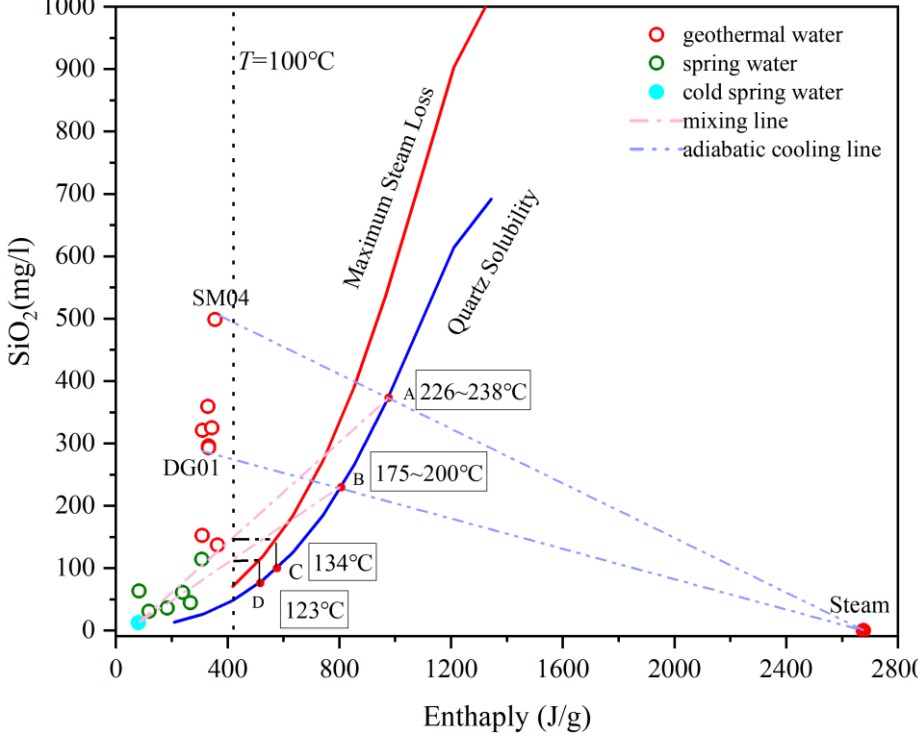

**Figure 6.** Silica-enthalpy mixing models of water samples along the TNR.

In summary, for T03-1, T03-2, T03, DG00, DG00, DG01, DG05, DG06, and SM04, where the sampling temperatures are close to the local boiling point, the reservoir temperatures estimated by the silica-enthalpy mixing models are close to those estimated by the quartz geothermometer at the time of maximum steam loss (Table 4). Quartz thermometers are more suitable than cationic ones for hot springs such as T01, T02, T04, T05, T06, and T07 (Figure 6)—where a large amount of shallow cold water is mixed during ascension towards the surface.

The circulation depths of water samples were calculated using to Equation (4):

$$H = h + (t - t0)/q \tag{4}$$

where $H$ is the circulation depth (km); $h$ is the depth of the constant temperature zone (km); $q$ is the geothermal gradient (°C/km) reflecting the geothermal change per one kilometer of the place below the constant temperature; $t$ is the reservoir temperature (°C); $t0$ is the temperature of the constant temperature zone (°C), namely the local average temperature. According to previous studies [21], the geothermal gradient in the study area is 45 °C/km. The depth of the constant temperature zone is about 20 m. The annual mean temperature in the TNR is 0.5 °C. The circulation depth of each sampling point is shown in Table 4.

### 4.4. Spatial Distribution of Springs and Earthquakes

To further understand the relationship between fault zones, hydrogeochemistry, and seismicity, this paper selects earthquakes with $M_L \geq 1$ in the study area since 1833. Figure 7 shows latitudinous plots of the variation of focal depth, the magnitude of the earthquake, and circulation depth. The seismic distribution in Figure 7a correlates with the spread of active regional faults and exhibits a clear sub-north–south zonation. The intensity of seismicity along the entire TNR is high, but earthquakes of magnitude 7 or greater occur only in the southern part of the rift (south of the IYS), indicating that the southern segment is more seismically active than the northern segment (north of the IYS) (Figure 7b). To investigate the difference in seismicity between the northern and southern segments of the TNR, this paper examines focal depth, circulation depth of hot sources, and $^3$He/$^4$He isotope data (from Klemperer et al. [11]) and plots them with latitude, respectively. The Indian continental plate is subducted beneath the Eurasian continental plate along the IYS, and the tremendous tectonic dynamics have resulted in extensive regional crustal deformation, shortening, and thickening in the Qinghai–Tibet region, controlling the seismic activity in the area [55]. Focal depths are mostly at shallow depths, primarily in the 15–35 km range (Figure 7c), which is consistent with the depth range of the region's low velocity, high conductivity layer [56]. This indicates that the occurrence of shallow earthquakes may be related to fluid activity in this layer and suggests that the low-velocity, high-conductivity layer is a possible seismogenic tectonic layer, i.e., a possible tectonic fracture zone or slip layer containing partial melting or free water in the crust. In terms of geochemistry, Figure 7d shows that the northern section of the TNR (SM04, DG00, DG01, DG05, DG06) is deeper than the southern section (T01, T02, T06, T07) for the depth of circulation of the hot springs. In addition, high values of the $^3$He/$^4$He isotope are concentrated north of the IYS (Figure 7e), with hot springs circulating to greater depths, with more mantle components and a stronger hydrothermal activity than that of the southern section.

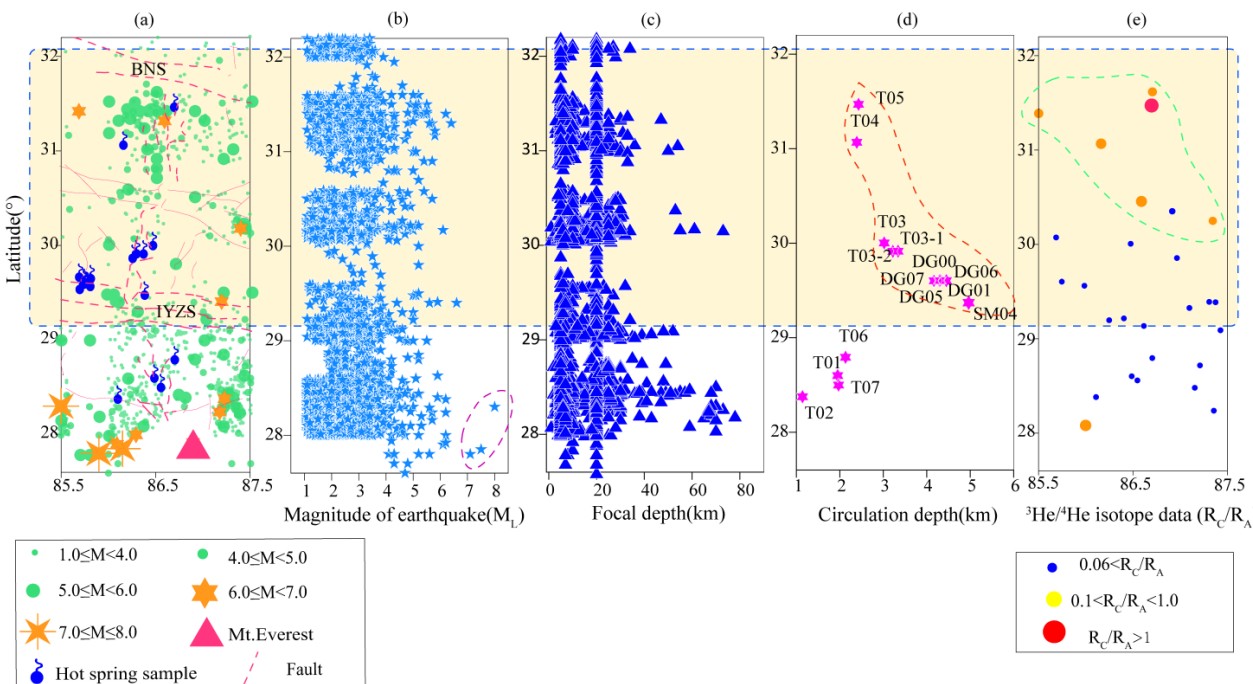

**Figure 7.** Spatial distribution of circulation depth of samples along the TNR and earthquakes: (**a**) the spatial disribution of spring and earthquake; (**b**) the magnitude of earthquake; (**c**) the spatial disribution of focal depth; (**d**) the circulation of springs; (**e**) the value of $^3$He/$^4$He isotope ($R_C/R_A$) and data from Klemperer et al. [11].

In the TNR's southern sector, the reservoir temperature is lower, circulation depth is shallower, hydrothermal activity is weaker, and the frequency of strong earthquakes is higher (three earthquakes with M > 7). In contrast, in the TNR's northern sector (the blue rectangle in Figure 7), the reservoir temperature is higher, circulation depth is deeper, hydrothermal activity is stronger, and the frequency of small and medium-sized earthquakes is higher. Tectonic dynamics fragment rocks, make faults more pronounced, and cause the frequent opening and closing of fractures, leading to strong seismic activity at the intersection of the fracture zone [11,57]. A higher fluid pore pressure in deep active fault systems drives fluid flow and reduces the effective positive pressure at the fracture surface, thereby weakening the fault strength and controlling seismic activity [58,59]. Additionally, a higher thermal reservoir temperature at the depth of the fault zone results in strong water–rock reactions that alter minerals and decompose them into low-friction clay minerals, thus changing the pressure and activity of the fault [60]. A greater circulation depth implies stronger water–rock reactions and more intense seismicity, indicating that subsurface fluids are perhaps involved to the seismogenic process. In the northern section of the TNR (where hot spring circulation is deeper), hydrothermal activity is stronger, and water–rock response at depth is stronger, fracture strength is weaker, and regional stresses are released in a relatively short period, resulting in a higher susceptibility to small earthquakes in this zone. Consequently, the geological and hydrogeochemical north–south segmentation of the TNR suggests that seismic activity in the southern section of the TNR is mainly influenced by tectonic activity, while the northern section is controlled not only by tectonic activity but also by strong hydrothermal activity in the area.

### 4.5. Temporal Variation of Springs and Earthquakes

Continuous measurements of major anions and cations have been taken every three days since September 2021 at the Yundong Spring (T06), which is situated near Mount Everest. Figure 8 illustrates the change in sodium, chloride, sulfate, and TDS over time for T06, and the standard deviation (σ) can be utilized to quantify variations from data.

Values above $2\sigma$ indicate significant positive anomalous variation, values below $2\sigma$ indicate significant negative anomalous variation. The blue bars represent small and medium earthquakes ($M_L \leq 3$) within 100 km, and large earthquakes ($M_L > 4$) within 400 km of T06. Small earthquakes near T06 are relatively concentrated (Figure 8a), and those with ML $\leq$ 4 did not significantly affect ion concentrations. The process of earthquake generation and occurrence involves stress loading and unloading, changes in the sub-stable and unstable state of fractures, as well as fluid exchange, transport, and hydro–rock reactions. Changes in regional tectonic stress can alter the groundwater circulation system (porosity, pressure, water flow, flow velocity, etc.), promote fluid exchange and transport, and re-establish water–rock chemical equilibrium. However, the effects of small-scale near-field seismic activity often has localized and limited effects on groundwater, and changes in the groundwater caused by moderately strong earthquakes are considered to be regional.

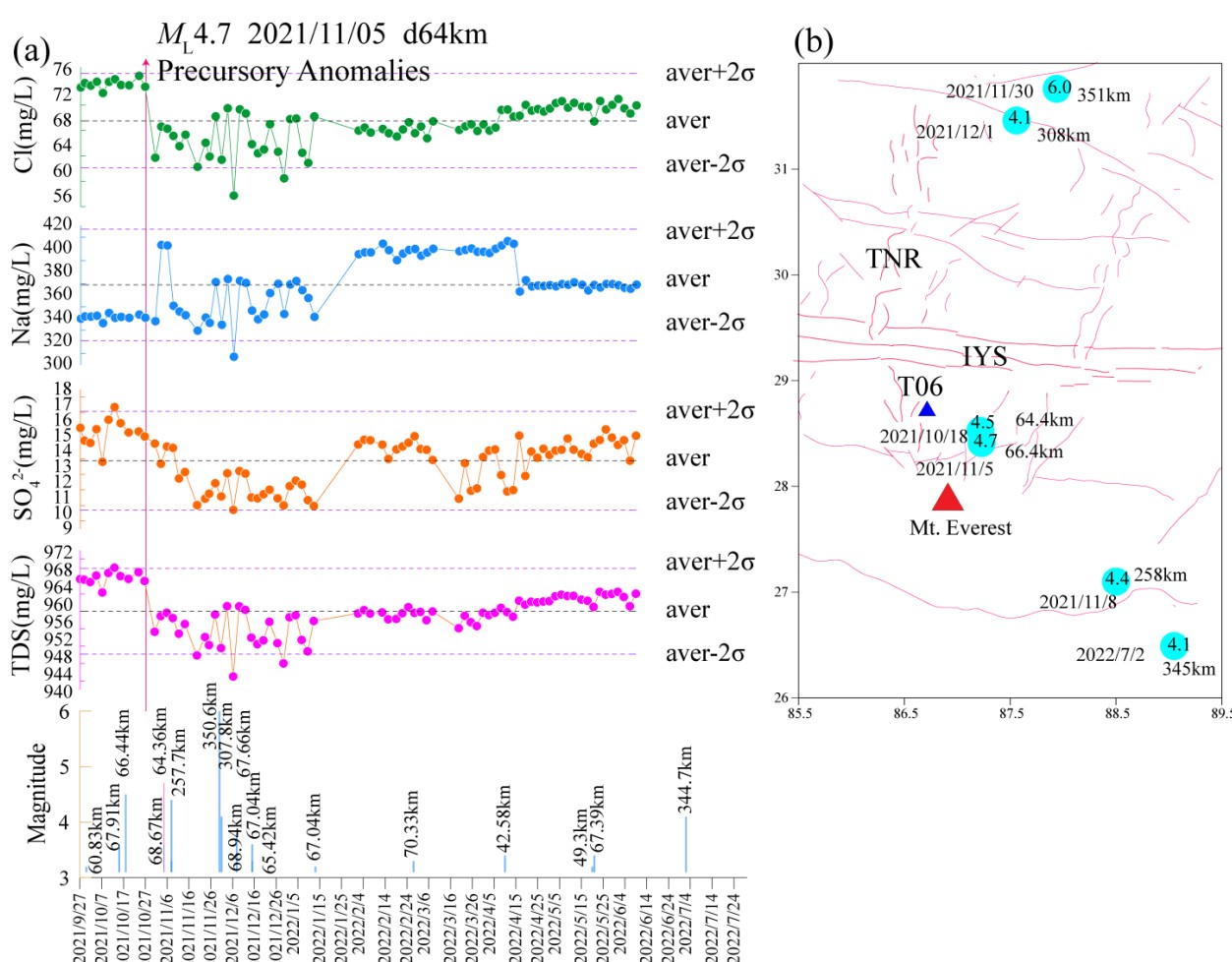

**Figure 8.** (**a**) The temporal variations in Na$^+$, Cl$^-$, SO$_4^{2-}$, TDS, and earthquakes; (**b**) the distribution plot of sampling T06 and earthquakes within 400 km and magnitude range over $M_L \geq 4.0$.

Eight days before an ML4.7 earthquake occurred 64.36 km away from T06 on November 5th, 2021 (red line), chloride and TDS concentrations dropped from their original high levels to negative anomaly zones. Many studies have focused on chloride ions and seismicity, as they are primarily derived from deep fluid upwelling and are less affected by shallow cold water mixing. Wang et al. [61] observed an abnormally high concentration of chlorine occurring 85 to 168 days before the *M*6.1 earthquake in Hualien and suggested that it may have had an association with the moderate earthquake in the Taiwan Strait. Li et al. [62] found that significant anomalies of high values of Na$^+$, Cl$^-$, SO$_4^{2-}$, and TDS

concentrations in hot springs might be pre-seismic anomalies of the Shuangbai $M_L$ 5.1 earthquake. Zhou et al. [6] found opposite changes in chloride concentrations in the northern and southern sections of the fault before the earthquake and concluded that the decrease in chloride concentration was relative to the locked state of the northern section of the fault at depth. Normally, under regional tectonic stress, chlorine at depth transports along the fracture to the shallow part, causing its concentration to increase. However, when the fracture is relatively locked, the fluid transports to the surrounding area, causing the monitored chlorine concentration to decrease. Meanwhile, chloride and sulfate, and TDS were always high before the $M_L$ 4.5 earthquake away 66.4 km from T06 on 18 October 2021, and no other changes were evident. The fact that T06 did not monitor changes in ion concentrations may be due to the small amount of data collected (only 8 sets of data before the earthquake) and the unknown trend of pre-seismic changes. This lack of change monitoring may also be connected to the orientation of the region stress field related to seismic events, where stress accumulation was not effectively transmitted to nearby faults [63]. A decrease in chloride concentration observed before the earthquake near T06 indicates a relatively locked fracture, resulting in the transport of fluid to the surrounding area and a subsequent decrease in chlorine concentration.

From the above analysis, the reaction of the hot springs to seismic events is regulated not only by the magnitude and epicenter distance but also by the tectonic stresses of the deep major fractures. Therefore, the continuous monitoring of hot springs near Mount Everest is crucial for the study of coupling relationships between hot springs, fractures, and earthquakes and for detecting earthquake precursory anomalies.

*4.6. Fluid Circulation Model*

In previous geological and geophysical studies conducted within southern Tibet's Indo-Asian continental collision context, materials with a significantly low resistance and low velocity were observed, which may be associated with the presence of partially molten magma in the upper crust [64,65]. Active fractures play a key role in transporting underground water and act as excellent reservoirs; they significantly control distribution of hot springs within rift system plane as well as temperature distribution/hydrothermal activity within profile [66]. Based on previous data and the research results in this paper, a fluid circulation model was established to explain the process of groundwater circulation migration in the TNR, as shown in Figure 9. Meteoric water infiltrates aquifers along fractures in the TNR and passes down the water-bearing zone into the deep. Deep water–rock interactions allow groundwater to convert into hot water carrying deep information, such as trace elements, gases, and so on, when the circulation depth rises to 1.2–5.02 km and the reservoir temperature exceeds 46.5–225.4 °C. Hot water then circulates upward along the hydraulic channels within fault zones due to pressure differences while mixing with shallow cold water during its ascent before finally emerging at the surface within valleys/river valleys at lower ground level.

Tectonic effects such as seismic incubation can alter the stress state of fracture zones, disrupt the original fluid–fault balance, and induce changes in fluid pressure, transport pathways, and the degree of water-rock reaction [60,67]. Consequently, the hydrological and geochemical characteristics of hot springs may also be affected. This paper investigates the hydrogeochemical characteristics of hot springs in the TNR and establishes a fluid circulation model, providing critical background data for earthquake tracking, fault activity, and groundwater circulation in the area.

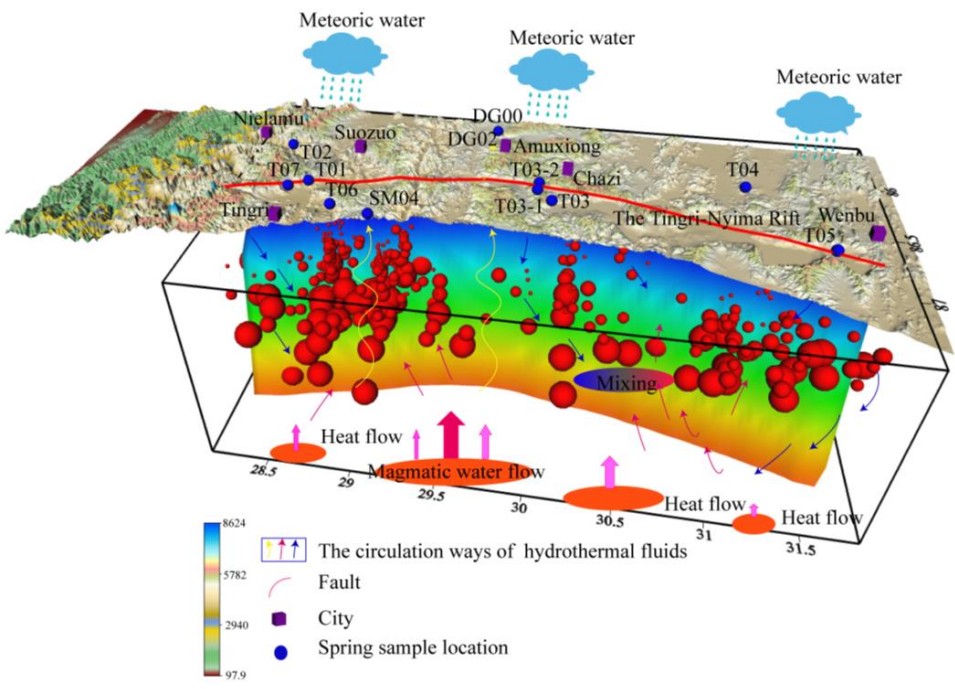

**Figure 9.** The fluid circulation model in the TNR.

## 5. Conclusions

Through the hydrogeochemical analysis of hot springs, this study provides valuable geological insights for monitoring earthquake precursors and investigating the interrelationship among fluids, fractures, and earthquakes in the region. Key findings revealed that the meteoric water from Mount Everest with a recharge elevation of approximately 7.5–8.4 km is the primary recharge for springs. The hydrochemical types of springs along the TNR are predominantly $HCO_3$-Na and Cl-Na. The geothermal reservoir temperature ranges from 46.5 to 225.4 °C, and the circulation depth ranges from 1.2 to 5.0 km. Deep hydrothermal activity and tectonic activity combine to control seismicity in the northern section of the TNR, while tectonic activity influences strong seismic activity in the southern section. Short-term seismic precursor anomalies of hydrochemical composition were detected eight days before the $M_L$ 4.7 earthquake that was 64.36 km away from T06, and it was controlled by tectonic stresses of the major fractures. Furthermore, a fluid circulation model was developed to describe the process of groundwater circulation migration. This study contributes towards filling the gap in hydrogeochemical investigations of the TNR and refines our understanding of fault activity, groundwater circulation, and seismic studies near the Mount Everest area. Further research is necessary to validate the key findings of this study.

**Author Contributions:** Methodology, J.S., Y.L., Y.Y. and X.R.; Software, J.T.; Formal analysis, M.H. and J.Z.; Data curation, F.L.; Writing—original draft, D.Z.; Writing—review & editing, Y.Z.; Supervision, X.Z. (Xiaocheng Zhou), X.Z. (Xiaoming Zhang), F.D. and Z.G.; Funding acquisition, X.Z. (Xiaocheng Zhou), Y.Y. and G.Q. All authors have read and agreed to the published version of the manuscript.

**Funding:** The work was funded by the Earthquake Tracking Task of CEA (Grant No. 2023010308), National Key Research and Development Project (2018YFE0109700, 2019YFC1509203), Central Public-interest Scientific Institution Basal Research Fund (CEAIEF2022030205, CEAIEF20220213, CEAIEF2022030200, CEAIEF20220507), and the National Natural Science Foundation of China (41673106, 42073063, 4193000170, U2039207), IGCP Project 724, and the Natural Science Foundation of Sichuan Province (2022NSFSC0210).

**Data Availability Statement:** All data can be obtained from the corresponding author by request.

**Acknowledgments:** The authors are grateful to the editors and reviewers for their constructive comments and suggestions.

**Conflicts of Interest:** The authors declare no conflict of interest.

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
