# Peer review of "Hydrogeochemical Study of Hot Springs along the Tingri—Nyima Rift: Relationship between Fluids and Earthquakes"

_water, doi:10.3390/w15081634_

Round 1
Reviewer 1 Report
The manuscript is an interesting work about the relationship between high-temperature fluids (thermal waters) and earthquakes in a little-studied area, north of Mount Everest. Despite being a good job, with adequate design and content, it is necessary to carry out a series of minor corrections:
1. line 21, indicates what T06 is (it is indicated later in the text, but it must be stated the first time it is cited).
2. It is necessary to improve the key of figure 1, especially of the symbols of Fig 2b and the acronyms.
3. Line 97, are you sure that it is a 0.45 meter membrane?
4. Table 1. It should be specified which samples are from the same location (eg T03 and T03-1?).
5. Table 2. Explain how the recharge altitude has been calculated, please.
6. lines 141-150. In explaining Figure 2, the possibility of the evaporation effect should be discussed.
7. Figure 5, include units in temperature, please.
8. Figure 6, the lines of the figure and their meaning are not well understood.
9. Table 3 is badly numbered (it puts table 2...).
Author Response
We feel great thanks for your professional review work on our article. As you are concerned, there are several problems that need to be addressed. According to your nice suggestions, we have made extensive corrections to our previous draft, the detailed corrections are in the attachment.

Reviewer 2 Report
Manuscript ID: water-2315086
Title: Hydrogeochemical Study of Hot Springs along the Tingri-Nyima Rift: Insights into the Relationship between Fluids and Earthquakes.
The comments are in the attached document.
Thank you so much for your attention.

Author Response

(The authors gave the same response as above.)

Reviewer 3 Report
The study contributes to the solution of an important practical problem for southern Tibet and other seismically active zones with similar conditions. The main results of the work were the identification of 8-day seismic precursors of the hydrochemical compositions of hot springs before earthquakes, as well as the creation of a conceptual model for groundwater origin and the process of hydrogeochemical cycling. As a suggestion, I propose the following. 1) The title can be shortened to “Hydrogeochemical Study of Hot Springs along the Tingri-Nyima Rift: Relationship between Fluids and Earthquakes”. 2) The figures often have a very small font that is difficult to read. It is necessary to increase its size and make a single size range of fonts for all figures. 3) The design of references in the text should be corrected: put a space before [No.], a number of references should be indicated in the form [10-12] or [10,11] instead of [10] [11] [12].
Author Response

(The authors gave the same response as above.)

Reviewer 4 Report
Dear authors,
I think the work is very interesting and it's very important. The problem is that when I was checking the raw data, the balances do not add up righly. There are errors in the anion-cation balance, the bicarbonates are very high and the conductivity is low for these ionic values.
Even there is some error in table 1, temperature (T03-3).
I think that before continuing with the review, the chemical data should be reviewed.
Author Response

(The authors gave the same response as above.)

Round 2
Reviewer 4 Report
Dear authors,
I consider the manuscript interesting and scientifically very important. It is difficult to read with the corrections and erasures
In my opinion it needs to improve in:
- deep revision of edition and English improvement
- the captions of figures are poor, not very informative and sometimes badly written
- the figures are small and it is difficult to distinguish the details
Moreover, I put some indications in the pdf manuscript.
Best regards.

Author Response

(The authors gave the same response as above.)
